# What Is the Evidence Base Linking Gender with Access to Forests and Use of Forest Resources for Food Security in Low- and Middle-Income Countries? A Systematic Evidence Map

**Ngolia Kimanzu** [1,2,*]**, Björn Schulte-Herbrüggen** [3]**, Jessica Clendenning** [4]**, Linley Chiwona-Karltun** [1]**, Kyla Krogseng** [1] **and Gillian Petrokofsky** [5]

1   Department of Urban & Rural Development, Swedish University of Agricultural Sciences, Box 7012, 750 07 Uppsala, Sweden; Linley.chiwona.karltun@slu.se (L.C.-K.); Kyla.Krogseng@slu.se (K.K.)
2   The Salvation Army, Box 5090, 102 42 Stockholm, Sweden
3   Stockholm Resilience Centre, Stockholm University, 106 91 Stockholm, Sweden; bsh@posteo.de
4   Center for International Forestry Research (CIFOR) Jalan, Bogor 16115, Indonesia; j.clendenning@cgiar.org
5   Oxford Long-Term Ecology Lab, Department of Zoology, University of Oxford, South Parks Road, Oxford OX1 3PS, UK; gillian.petrokofsky@zoo.ox.ac.uk
*   Correspondence: Ngolia.kimanzu@fralsningsarmen.se

**Abstract:** In nearly all parts of the world, an important part of people's livelihood is derived from natural resources. Gender is considered one of the most important determinants of access and control over forests. It is thought that women and men within households and communities have different opportunities and different roles and responsibilities in relation to forest use. It is probable that when women have equal access to forests, better food security outcomes can be achieved for individuals and households that are dependent on forests for their livelihoods. A systematic evidence map of the evidence base linking gender with access to forests and use of forest resources for food security was undertaken. Ten bibliographic databases and 22 websites of international development and conservation organisations were searched using keywords suggested by stakeholders. Other articles were found by emailing authors and organisations to send potentially relevant publications. 19,500 articles were retrieved from bibliographic databases and 1281 from other sources. After iterative screening, 77 studies were included: 41 focussed on Africa, 22 on Asia, 12 on Latin America, 2 were global. Most indicators of food security measure access to food, measured by total consumption, expenditure, or income. Studies showed strong gender specialisation: commercial access and utilisation of forests and forest products dominated by men, whereas access for subsistence and household consumption is almost exclusively the task of women. Despite the large number of studies reviewed, limitations of the evidence base, including methodological heterogeneity, a dominance of case studies as the study design, and unequal geographical representation in study locations, make it difficult to generalise about the overall importance of gender and its effect on access to and use of forests for food security in developing countries. The critical gaps in the evidence base include geographical representation in primary research and a greater breadth of study designs to assess gender implications of access to forest resources globally.

**Keywords:** access; equity; forests; gender relations; income; livelihoods; non-timber forest products



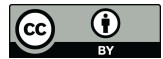

## 1. Background

More than 1.6 billion people are believed to rely on forests for their livelihood in one way or another. Another 60 million indigenous people rely on woods almost entirely [1]. Nearly half of the world's population, or those who live on $2 or less per day, is directly supported by the world's forests [2], which provide resources that operate as safety nets for their livelihoods, such as shelter, food, and fuel wood [3–5]. Further, the World Bank claims that "sustainable use of forests requires the participation of all rural populations,

including women" [1]. A number of research and publications over the last two decades have linked gender inequities in access to forest resources to higher levels of poverty [6–9]. Women's demands and priorities, particularly in connection to natural resources, can easily be disregarded owing to established socio-cultural practices and gendered power dynamics [10,11]. Understanding the role of gender relations in development studies is perhaps more important now than ever in the light of SDG 5 gender Equality—"achieve gender equality and empower all women and girls", and SGD 2—"zero hunger" (https://www.un.org/sustainabledevelopment, accessed on 12 July 2020).

In this systematic map, we define "gender" as a set of social constructs that attribute different attitudes, abilities, personality traits, and behavioural patterns to men and women, as well as systems of differentiation and power, which are evident in an unequal division of resources and labour between men and women. In low-income rural areas in particular, gender relations can define how men and women access forest resources and can place a disproportionately large burden on women to manage the household, including food sourcing responsibilities, compared to men.

According to Ribot and Peluso [8], an understanding of access should extend beyond property rights as previously proposed by Schlager and Ostrom [12]. Access should be seen as the absolute right to enter a given space or physical property. In this systematic map, we conceive access as "the ability to benefit from things in and from the forest". We also define access as encompassing physical dimensions, such as distance from the forest and legal or other rights of entry into the forest.

In rural areas of low- and middle-income countries, women are frequently in charge of household cooking and firewood collection [13]. According to several studies, women are better educated about medicinal plants and other non-timber forest products (NTFP) than men [14–16].

Other research on food security outcomes has also revealed how crucial women are in providing nutritious diets, particularly when it comes to forest resources. [17]. This is an important consideration during the agricultural lean season, since it might seriously affect a household's food security [2]. In this systematic map, we adopt the definition of food security of the World Food Summit's Rome Declaration: "Food security exists when all people, at all times, have physical and economic access to sufficient, safe and nutritious food that meets their dietary needs and food preferences for an active and healthy life" [18]. This was further developed by Arnold et al. in the context of forests [3]. Access to food is insufficient in and of itself, as age, gender, and culture may all play important roles. Forests make an important contribution to diets, particularly in rural communities, and depending on the type of forest produce used, by (a) increasing the diversity of diets, particularly during the agricultural lean season; (b) supplying supplementary calories as snacks or complementary food items; and (c) providing employment for people in towns and rural areas [17,19]. NTFPs may help to alleviate various nutritional and micronutrient deficits, depending on the precise combination of foods [20].

In the absence of a systematic evaluation of all the available evidence of relevance to these issues, we undertook a systematic map following guidelines for the conduct of systematic evidence synthesis [21].

## 1.1. Objective of the Map

The objectives of the systematic evidence evaluation was to assemble and analyse the evidence base on linkages between gender and forest access and use of forest resources for food security in low- and middle-income countries (https://datahelpdesk.worldbank.org/knowledgebase/articles/906519-world-bank-country-and-lending-groups, accessed on 12 July 2020).

## 1.2. Stakeholder Workshop

A stakeholder workshop was held in Bogor, Indonesia, from 18 to 21 February 2014 to discuss the current knowledge base on how gender mediates access to and use of

forest resources. Four of the current authors participated in that workshop (NK, JC, LCK, GP), together with eight other academics and policymakers in the field of gender studies. All 12 people contributed to a protocol that defines the method used in the current paper [22]. The protocol provides more information about the stakeholder workshop. The workshop concluded with the decision to undertake a systematic map to assess the extent of the evidence base of relevance to the questions we posed. Stakeholders agreed to use a systematic map methodology to assess the geographic distribution of relevant research to describe the characteristics of the research and the outcomes reported (e.g., food and income security). Systematic maps involve reviewing a large body of literature to define review questions. We, therefore, chose a systematic map because we expected the knowledge base to be insufficient for a full systematic review and meta-analysis, and a systematic map could provide the foundation for further research by identifying what is already known. One of the important aims of the evidence synthesis was to highlight knowledge gaps in research linking gender with forest access and food security.

*1.3. Research Questions*

The primary research question is: What is the evidence that gender affects access to and use of forest assets for food security?

The sub-questions of the map are:

i.  What is the evidence that women's access to forest resources (or assets) improves household food security compared to that of men?
ii. What does the evidence show as gender disparities in access to and use of forests?

*1.4. Elements of the Review Question*

Adapting the commonly used population, exposure, control, and outcomes (PECO) framework to a framework better suited to our question, we formulated a framework based on subject, exposure, and outcomes (SEO). Table 1 shows the elements of this framework.

**Table 1.** Elements of the systematic map question.

| Subject | Forest resources and assets in low- and middle-income countries (as defined by the World Bank, 2014) |
| --- | --- |
| Exposure | Women- or female-headed households that access and use forest resources and assets |
| Outcomes | Changes in food security, defined by a range of indicators |

## 2. Methods

*2.1. Search Strategy*

Searches were carried out in the following bibliographic databases and aggregators in January–March 2017:

- AGRIS; (1974–current) (www.agris.fao.org)
- CAB Abstracts (1910–current, accessed through Web of Science)
- Google (www.google.com)
- Google Scholar (scholar.google.com)
- JSTOR (www.jstor.org)
- ProQuest Dissertations & Theses (1995–current) (www.proquest.com)
- MEDLINE Opens (1950–current, accessed through Web of Science)
- Scopus (http://www.scopus.com) (1823–current)
- Web of Science Core Collection (1945–current)(www.wokinfo.com)
- Zoological Record (1990–current, accessed through Web of Science)

In addition, grey literature was searched in a large number of institutional websites, suggested by stakeholders (see Supplementary Materials—Table S1)

## 2.2. Search Terms and Languages

The search terms were suggested during the stakeholder workshop and augmented through iterative exploration using QSR Nvivo 11 software (published by QSR International, https://www.qsrinternational.com/nvivo-qualitative-data-analysis-software/home, accessed on 12 July 2020) to arrive at a final set of terms that successfully retrieved the test library of 20 articles of known relevance to the study (Supplementary Materials—Table S2). The searches were carried out using the English terms listed in Table 2, structured around the SEO framework (Table 1). Keywords were first connected using Boolean operator OR within each column and then using Boolean operator AND across columns. The asterisk (*) indicates truncation of a search term, e.g., the term "forest*" will search "forest", "forests", "forestry", etc. It is a common convention in bibliographic database searching.

**Table 2.** Terms used to search for relevant articles.

| Subject | Exposure | Outcomes |
|---|---|---|
| (forest *; tree *; agroforest *; woodland; mangrove; savanna *; shrub; wood; bush; "rights to land"; biodiversity) | (Gender; "female headed"; "male headed"; "sexual roles"; "role conflicts"; "woman's status"; "women's rights"; "man's status"; "men's rights"; "sexual discrimination"; household *; widow) | ("food security"; income; cash; wealth; poverty; hunger; nutrition *; malnutrition; vitamin *; diet; livelihood *; rights; diversity; consumption; equity) |
| AND | AND | |
| [List of 150 low- and middle-income countries—see Supplementary Materials—Table S3 for details] | (Labour *; "cash crop"; tenure; "tenure system *"; "land tenure"; "agricultural tenure"; "agricultural households"; nonfarm; property; forage *; "staple food"; "land rights"; asset *; resource *; bushmeat; fuelwood; firewood; charcoal; vegetable; plant; fruit; mushroom; timber; honey; access; "forest product"; NTFP; participatory; education) | |

The asterisk (*) indicates truncation of a search term.

The final search strings used in Web of Science and Scopus are shown in Supplementary Materials—Table S4. The search string was adapted slightly to accommodate features of individual database and simplified for Google and Google Scholar to accommodate their less sophisticated search capability (Supplementary Materials—Table S4).

## 2.3. Article Screening and Study Inclusion Criteria

Articles retrieved were screened sequentially for relevance at the title, abstract, and full-text stages by two reviewers (NK and BSH), who checked their common understanding of the inclusion criteria using random samples of 100 articles and calculating their kappa scores. They engaged in iterative testing and discussion of differences of opinion on inclusion until reaching a kappa score greater than 0.6.

## 2.4. Study Inclusion Criteria

To be included in the systematic map, a study had to meet all the inclusion criteria:

Subject—study reports how women or men use or access forest resources in low- and middle-income countries. All natural and planted forest types were included.

Exposure—study reports how women- or female-headed households access and use the forest and its resources.

Outcome—study reports an outcome (or effect) related to food security.

Study designs can include quantitative and qualitative studies, which are not limited to peer-reviewed journals. Secondary publications, e.g., literature reviews, were assessed for background and contextual purposes, but were not included in the systematic map.

### 2.5. Study Quality Assessment

Guidelines for systematic maps state that study quality assessment is not required, but we were interested in assessing the quality of the evidence base if possible. There is no single method that is appropriate for all types of literature contained in systematic reviews or systematic maps. We trialled the use of the environmental risk of bias tool as recommended by Bilotta et al. [23] and an extensively cited hierarchy of study designs from the social sciences [24,25] using a template with a summary explanation of the assigned ratings (Supplementary Materials—Table S5. We tested the template using studies in the test library (Supplementary Materials—Table S2).

### 2.6. Data Extraction and Coding

A data coding and extraction template was agreed by stakeholders and authors and tested by coders. Table 3 shows the metadata that were coded and extracted. Supplementary Materials—Table S5 shows the data extraction and coding template used.

**Table 3.** Data coding and data extraction elements.

| Article Details (Nature of Evidence) | ID |
|---|---|
| | Citation |
| | Publication type |
| | Source type |
| | Year of publication |
| | Location of primary author (country) |
| Coverage of evidence | Location of data collection (country) |
| | Coastal (coastal = up to 10 km from coast or large lake)/not coastal |
| | Elevation |
| | Rainfall (annual, mm) |
| | Rural/urban |
| | Forest type accessed |
| Study design | Qualitative or quantitative |
| | Method |
| | Scale of study (local, regional or national, international) |
| | Number of study sites in analysis |
| | Sample size at each study site (e.g., number of interviews or focus groups) |
| | Sample unit (person, household, focus group) |
| | Duration of data collection |
| Access to forest and markets | Distance to forest |
| | Market access |
| | Main NTFP accessed |

**Table 3.** *Cont.*

| Article Details (Nature of Evidence) | ID |
| --- | --- |
|  | Forest tenure |
| Study details (food security outcomes) | Is food security assessed? |
|  | Is nutrition security assessed? |
|  | Is income assessed? |
|  | Food security outcome through specific engagement |
| Gender effects | Positive, neutral, negative effects |
| Notes | Details of gender effects (extracts from primary studies) |
| Critical appraisal | QUALITATIVE studies rating |
|  | QUANTITATIVE studies rating |
|  | Bilotta's risk of bias (for all studies) |

Fully coded data files and an interactive, online map of all included studies were prepared to report the evidence base discovered through the systematic methods employed. The interactive map was based on an open-source tool developed by Dr Andrew Martin of the University of Oxford. The map can be filtered to explore variables of interest to the review question.

## 3. Results

All included studies, with details of data extracted and coding applied, are shown in Supplementary Materials—Table S5. An interactive, online map of all is available, which includes coding and data extractions for all included studies, with full bibliographic details. The map can be filtered to explore variables of interest to the review question—see https://oxsrev.github.io/evidencemaps/gender/, accessed on 12 July 2020.

### 3.1. Search Results

Initially 19,500 studies were identified through database searching. In addition, 1281 studies were identified through other sources. After removing duplicates, we screened 14,005 articles by title and abstract. Initial search terms such as gender, forests, and food security led to a high percentage of irrelevant results. This led to the exclusion of 13,379 articles at the title and/or abstract level. For full-text screening, we were able to identify 625 studies across all the different sources. A total of 77 studies met the inclusion criteria. (Figure 1).

We summarise the results obtained from data extraction and coding of all 77 included studies below in a narrative summary of the evidence base. A comprehensive Excel file of the data and codes applied to each individual study is available in Supplementary Materials—Table S5.

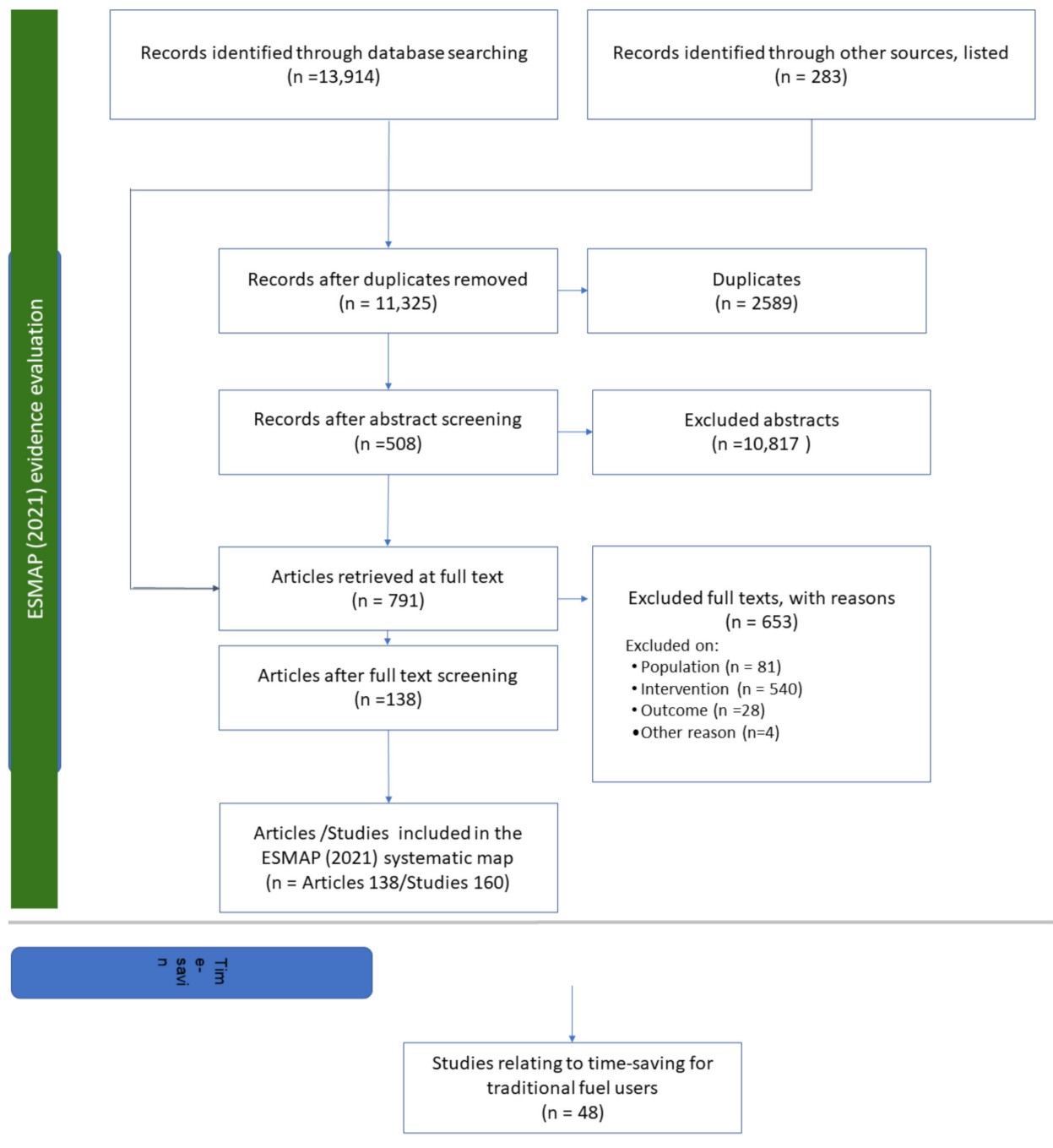

**Figure 1.** Flowchart of screening decisions (adapted from Haddaway et al. [26]).

### 3.2. Year of Publication

Figure 2 shows the year of publication of each article in the evidence map, with a generally increasing volume of articles over time, in common with scientific publication.

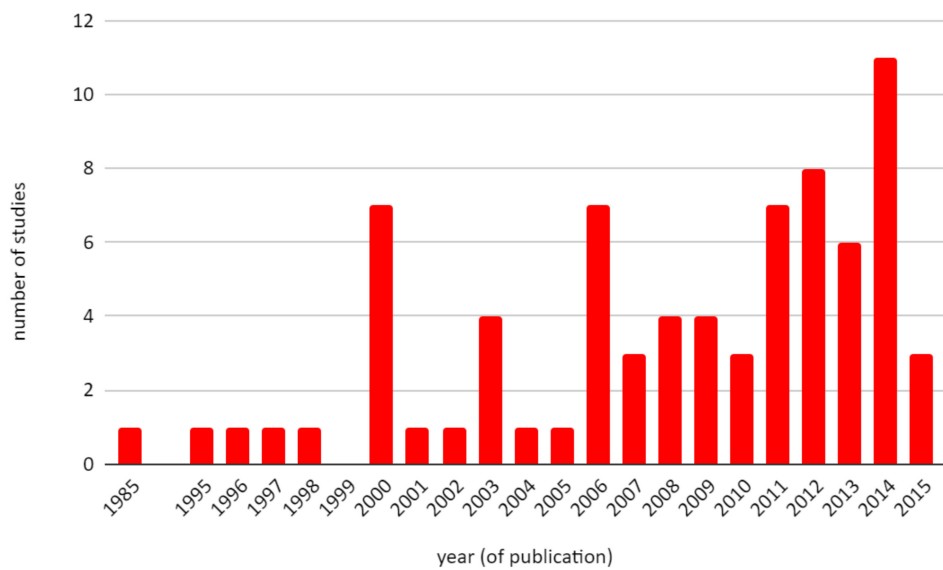

**Figure 2.** Number of studies by year of publication.

### 3.3. Location of First Author's Institution

We captured the address of the first author of each included study to give an indication of where research relating to gender, food security, and forest access is occurring, Table 4 shows that 60% of the research publications originated in low-and middle-income countries and 40% in high-income countries.

**Table 4.** Location of first author.

| | | | |
|---|---|---|---|
| Ethiopia | 7 | Netherlands | 2 |
| India | 6 | Sri Lanka | 2 |
| USA | 7 | Bhutan | 1 |
| South Africa | 5 | Brazil | 1 |
| Nigeria | 4 | Burkina Faso * | 1 |
| UK | 5 | Cambodia | 1 |
| Canada | 3 | Ghana * | 1 |
| Denmark | 3 | Japan | 1 |
| Germany | 3 | Kenya | 1 |
| Norway | 3 | Malawi | 1 |
| Sweden | 3 | Malaysia | 1 |
| Vietnam | 3 | Tanzania | 1 |
| Benin | 2 | Turkey | 1 |
| Bolivia | 2 | Uganda | 1 |
| Indonesia | 2 | Zambia | 1 |
| Mexico | 2 | NA | 1 |

* Joint institution.

### 3.4. Location of Studies

(a) Country where data were collected is shown in Figure 3.

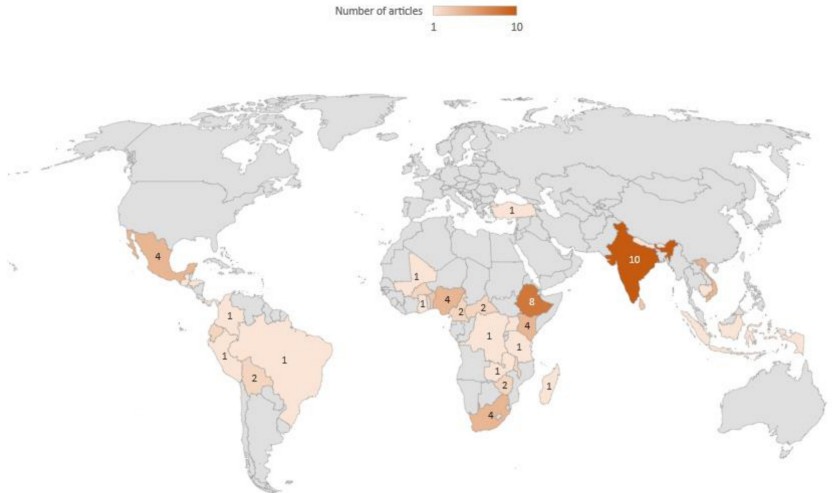

**Figure 3.** Location of study sites. (Copyright Australian Bureau of Statistics).

(b)     Whether study site was coastal or not (coastal = up to 10 km from coast or large lake) is shown in Figure 4.

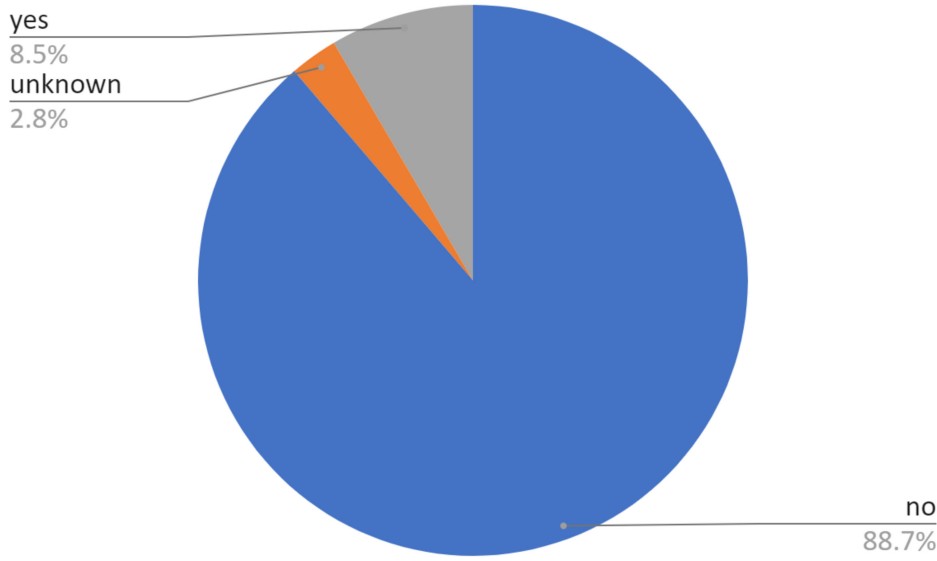

**Figure 4.** Coastal/non-coastal location of study sites.

(c)     Elevation of study sites

Studies were grouped into those at low elevation (below 500 m), mid-elevation (500–1000 m), and high (above 1000 m). Where sites had a range of elevations, they are recorded with the highest value (Figure 5).

(d)     Rainfall at study sites

Almost half the studies (31) did not report rainfall; of those that did, the range was very wide: from 200 mm/year to over 5000 mm/year.

(e)     Urban/rural study sites

The vast majority of studies were from rural sites (72), with only 3 from urban (or urban/peri-urban or rural/urban).

(f)     Details of forests and access to forest

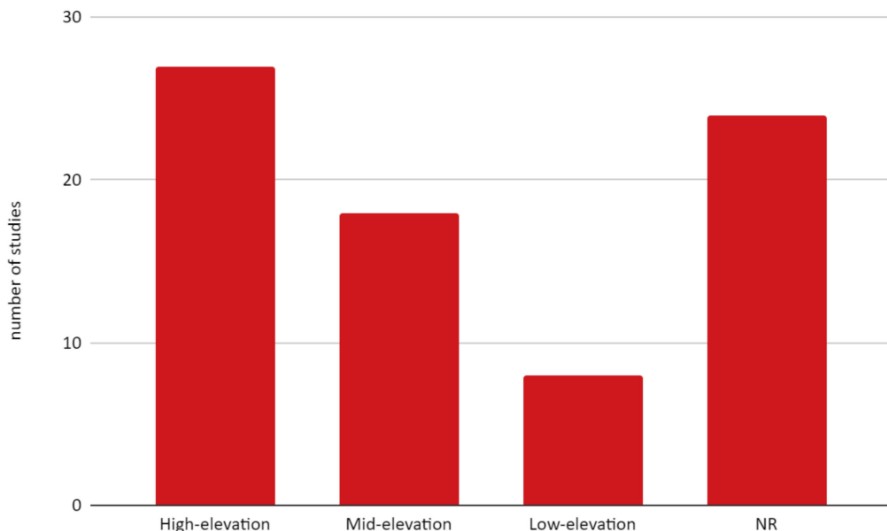

**Figure 5.** Elevation of study sites.

The majority of forests accessed were reported as "natural forest" (68 studies), with 4 natural/plantation, 1 plantation, and 4 not reported. The study sites were "adjacent to forest" (63 studies), with 3 inside the forest, 2 at some distance from the forest, and 8 not reporting.

(g) Market access

Market access was mostly low (no road or market far away), or very low (no road and market far)—23 and 7 studies, respectively, with only 18 studies reporting good access. Percentage studies had mixed, good/low results, and 24 did not report data.

*3.5. Governance of the Forest*

This was reported as state-owned, commons, traditional land tenure, private, or mixes of different regimes (Figure 6).

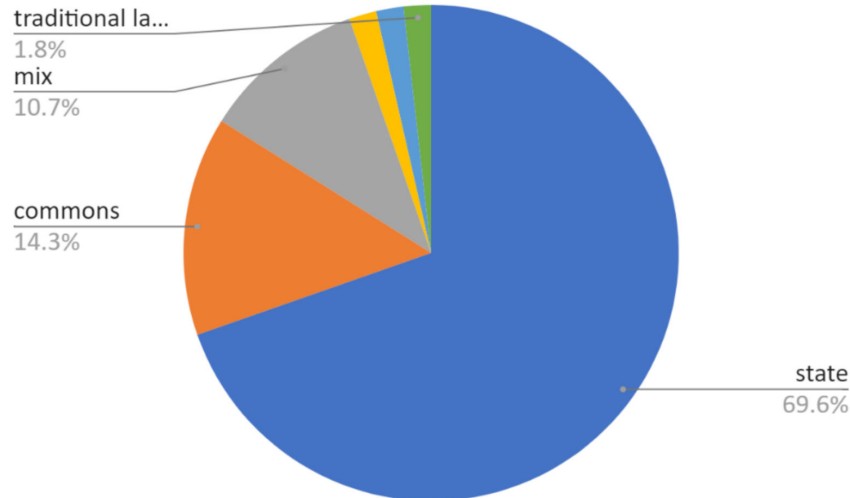

**Figure 6.** Forest tenure.

*3.6. Study Design*

(a) Number of study sites in the analysis

The number of sites included in the analysis of included studies ranged between 2 and 333, with a mode of 2 and median of 4 (Figure 7).

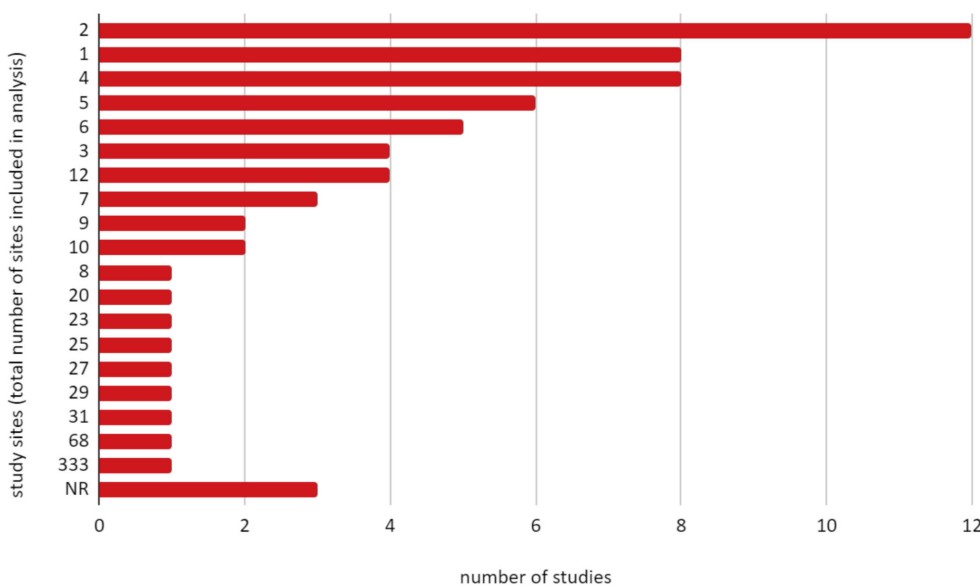

**Figure 7.** Number of sites included in the analysis.

(b)    Sample unit and size

Most studies were conducted at the unit of household (Table 5); the sample size at each study site (e.g., number of interviews or focus groups) ranged from 8 to 8094, with a mean of 165.7692308, median 141, mode 32, and 2 not recorded.

**Table 5.** Unit of analysis.

| Sample Unit | Number of Studies |
| --- | --- |
| Households | 58 |
| Individuals | 13 |
| NR | 6 |
| Focus group | 2 |

(c)    Duration of data collection

Data were collected between 1 and 36 months, with mean of 11.9, median 11, mode 12, and 17 studies not reporting duration of data collection (Figure 8).

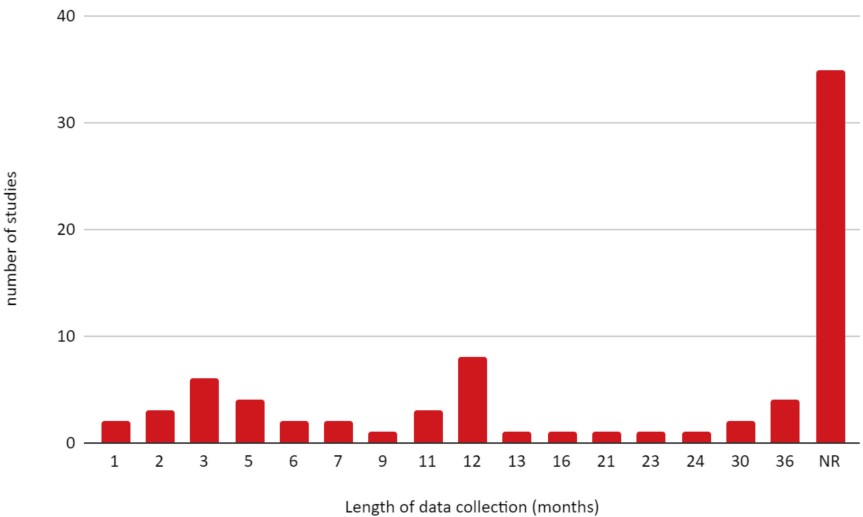

**Figure 8.** Duration of data collection.

### 3.7. Forest Resources and Food Security, Nutrition Security, and Income

Most studies reported income (72 studies) as a proxy for food security. Food and nutrition security was reported by only 43 and 45 studies, respectively (Figure 9). Food security was achieved through trading (9 studies), harvesting (6 studies), consumption (1 study), or, mostly, a mix of harvesting and processing (54 studies); however, 9 studies did not specify how food security was achieved. Most studies (63) reported a mix of non-timber forest products (NTFP), 10 studies reported wild food as the main NTFP, 1 reported charcoal/firewood, and 1 reported brooms as the main source of NTFP.

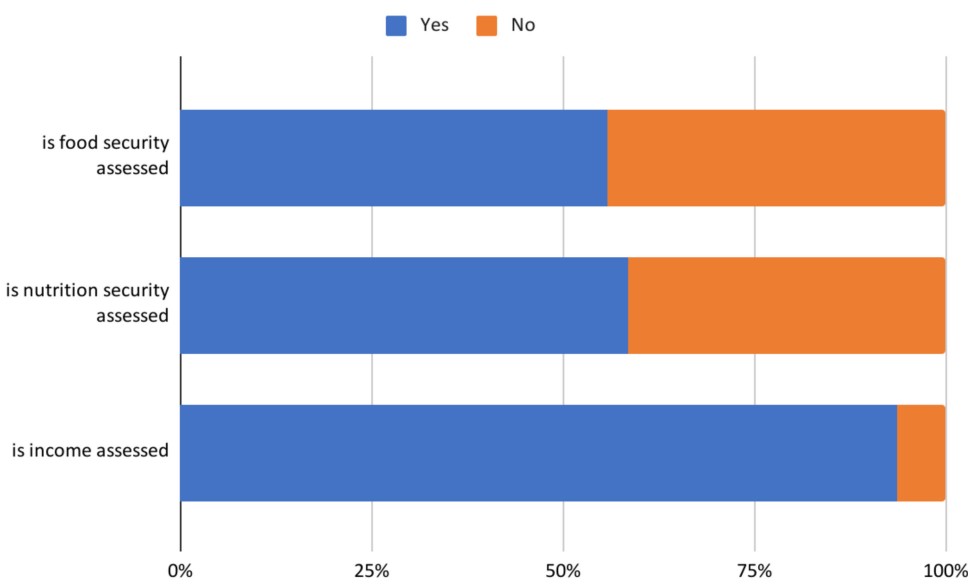

**Figure 9.** Food and nutrition security, and income.

Food security outcomes were achieved mostly by a mix of engagements (54 studies), with 9 studies reporting trading, 7 reporting harvesting, 1 processing, and 6 unspecified.

### 3.8. Research Design

Studies were mostly quantitative (30) or mixed methods (44), with only 3 qualitative studies (Figure 10).

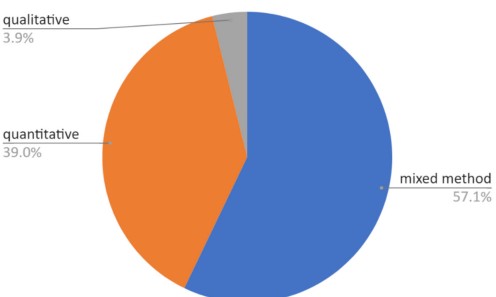

**Figure 10.** Research design of studies.

The main difference in the included studies is in their design. Thirty-one studies use regression models, with income as a response variable, whereas household head, gender, education level, age, family size, and distance to market were some of the main explanatory variables [27–56]. Six studies, without employing the rigour of regression, ran analysis of variance (ANOVA) to compare the socio-economic profiles and income of households derived from the access and utilisation of forest products [57–62]. Studies using other research designs are included in the systematic map [63–103].

*3.9. Impact of Gender*

We did not undertake effect size analysis, and the observations below are taken from author reporting of their data. We are aware of the danger of "vote counting" and make no further assessment based on numbers reporting positive/negative or neutral effects. However, it is useful to summarise the state of the evidence base for this systematic map and recommend future metadata analysis of subsets that we present.

The articles were categorised as having a positive gender effect when they reported that women had significantly better access to forests than men and derived food security directly through consumption of forest products or indirectly through income generated by sale of forest products. When articles reported that men derived significantly better access to forests and derived food security directly through consumption of forests products or indirectly through sale of forest products, we categorised such articles as having a negative effect on gender. Some articles, however, reported no significant differences between men and women in their access to forests for food security, and we categorised these as having a neutral gender effect. There were 22 studies that showed strongly positive or positive effects on gender, 41 that showed neither positive nor negative, or mixed results, and 14 that showed negative effects. Table 6 shows the studies, together with details of the country of the study, the setting of the studies, market access, and forest tenure. It also serves to document all the studies included in the systematic map. The interactive map and the associated extraction table (Supplementary Materials—Table S5) show details of the text extracts relating to gender effects on food security.

**Table 6.** Studies reporting positive, neutral/mixed, or negative impact of gender.

| Study | Gender Effect | Country | Rural/Urban Setting | Market Access | Forest Tenure |
|---|---|---|---|---|---|
| Angelsen et al. (2014) [27] | Positive | Global | Rural | NR | NR |
| Babulo et al. (2008) [28] | Positive | Ethiopia | Rural | NR | NR |
| Das (2011) [29] | Positive | India | Unknown | NR | NR |
| Heubach et al. (2011) [30] | Positive | Benin | Rural | Low | Traditional Land tenure |
| Lybbert et al. (2010) [31] | Positive | | | | |
| Mbuvi & Boon (2009 [32] | Positive | Kenya | Rural | Good | State |
| Misra & Dash (2000) [33] | Positive | India | Rural | Good (road and market close) | |
| Narayanan & Kumar (2007) [34] | Positive | India | Rural | NR | State |
| Noss & Hewlett (2001) [35] | Positive | Central African Republic | Rural | Very low (no road and market far) | State |
| Odebode (2005) [36] | Positive | Nigeria | Rural | Good (road and market close) | State |
| Ogle et al. (2003) [37] | Positive | Vietnam | Rural | NR | State |
| Padmanabhan (2011) [38] | Positive | India | Rural | NR | State |
| Pouliot (2012) [39] | Positive | Burkina Faso | Rural | NR | State |
| Shackleton & Campbell (2007) [40] | Positive | South Africa | Rural | Good (road and market close) | State |
| Shumsky et al. (2014) [41] | Positive | Kenya | Rural | Good (road and market close) | State |
| Sinclair & Ham (2000) [42] | Positive | India | Rural | Very low (no road and market far) | State |
| Singh et al. (2015) [43] | Positive | India | Rural | NR | State |

**Table 6.** *Cont.*

| Study | Gender Effect | Country | Rural/Urban Setting | Market Access | Forest Tenure |
|---|---|---|---|---|---|
| Uzokwe (2014) [44] | Positive | Nigeria | Rural | Low (no road or market far) | |
| van Dijk et al. (2003) [45] | Positive | Cameroon | Rural | Low (no road or market far) | |
| Vazquez-Garcia (2008) [46] | Positive | Mexico | Rural | Low (no road or market far) | State |
| Worku et al. (2011) [47] | Positive | Ethiopia | Rural | Low (no road or market far) | |
| Yusuf et al. (2013) [48] | Positive | Ethiopia | Rural | Low (no road or market far) | |
| Asfaw et al. (2013) [49] | Neutral/mixed | Ethiopia | Rural | Good | NR |
| Chukwuone & Okeke (2012) [50] | Neutral/mixed | Nigeria | Rural | NR | NR |
| Gatiso & Wossen (2015) [51] | Neutral/mixed | Ethiopia | Rural | Good | Commons |
| Hegde & Enters (2000) [52] | Neutral/mixed | India | Rural | Good (road and market close) | NR |
| Jones et al. (2006) [53] | Neutral/mixed | Madagascar | Rural | Mixed | NR |
| Kabubo-Mariara (2013). [54] | Neutral/mixed | Kenya | Rural | NR | Mixed |
| Kamanga et al. (2009). [55] | Neutral/mixed | Malawi | Rural | Mixed | NR |
| Madge (1995). [56] | Neutral/mixed | Gambia | Rural | NR | Mixed |
| Marshall & Newton (2003). [57] | Neutral/mixed | Mexico | Rural | Mixed | Mixed |
| Martin del Campo-Hermosillo (2010). [58] | Neutral/mixed | Mexico | Urban-periurban | Good | Commons |
| Mishra & Chaudhury (2012). [59] | Neutral/mixed | India | Rural | Good (road and market close) | Commons |
| Morsello et al. (2012). [60] | Neutral/mixed | Brazil and Bolivia | Rural | | State |
| Mujawamariya & Karimov (2014). [61] | Neutral/mixed | Kenya | Rural | NR | Commons |
| Nesheim & Stoelen (2012). [62] | Neutral/mixed | Guatemala | Rural | Very low (no road and market far) | State |
| Ojo et al. (2013). [63] | Neutral/mixed | Nigeria | Rural | NR | State |
| Ostwald & Baral (2000). [64] | Neutral/mixed | India | Rural | NR | State |
| Pouliot & Treue (2013). [65] | Neutral/mixed | Ghana and Burkina Faso | Rural | NR | State |
| Powell & Johns (2011). [66] | Neutral/mixed | Tanzania | Rural | Low (no road or market far) | State |
| Quinonez-Martinez et al. (2014). [67] | Neutral/mixed | Mexico | Urban | Good (road and market close) | State |
| Remis & Jost Robinson (2014). [68] | Neutral/mixed | Central African Republic | Rural | Low (no road or market far) | State |
| Shackleton (2004). [69] | Neutral/mixed | South Africa | Rural | Good (road and market close) | State |
| Shackleton et al. (2002). [70] | Neutral/mixed | South Africa | Rural | Good (road and market close) | State |

**Table 6.** *Cont.*

| Study | Gender Effect | Country | Rural/Urban Setting | Market Access | Forest Tenure |
|---|---|---|---|---|---|
| Shams & Ahmed (2000). [71] | Neutral/mixed | Cambodia | Rural | Good (road and market close) | State |
| Sharaunga et al. (2013). [72] | Neutral/mixed | South Africa | Rural | Good (road and market close) | State |
| Shrestha & Dhillion (2006). [73] | Neutral/mixed | Nepal | Rural | NR | State |
| Singh et al. (1985). [74] | Neutral/mixed | India | Rural | NR | State |
| Siren & Machoa (2008). [75] | Neutral/mixed | Equador | Rural | Very low (no road and market far) | State |
| Sunderland et al. (2014). [76] | Neutral/mixed | Global | Rural–urban | Good (road and market close) | Mixed |
| Tadesse et al. (2014). [77] | Neutral/mixed | Ethiopia | Rural | Low (no road or market far) | State |
| Toksoy & Alkan (2010). [78] | Neutral/mixed | Turkey | Rural | Very low (no road and market far) | State |
| Uberhuaga et al. (2012). [79] | Neutral/mixed | Bolivia | Rural | Low (no road or market far) | State |
| Van Hoang et al. (2008). [80] | Neutral/mixed | Vietnam | Rural | Low (no road or market far) | |
| Velasquez Runk et al. (2007). [81] | Neutral/mixed | Panama | Rural | Low (no road or market far) | State |
| Wickramasinghe (1997). [82] | Neutral/mixed | Sri Lanka | Rural | Low (no road or market far) | State |
| Wickramasinghe et al. (1996). [83] | Neutral/mixed | Sri Lanka | Rural | Low (no road or market far) | |
| Viet Quang & Nam Anh (2006). [84] | Neutral/mixed | Vietnam | Rural | Low (no road or market far) | |
| Vodouhe et al. (2011). [85] | Neutral/mixed | Benin | Rural | Low (no road or market far) | State |
| Vodouhe et al. (2009). [86] | Neutral/mixed | Benin | Rural | Low (no road or market far) | |
| Wong & Godoy (2003). [87] | Neutral/mixed | Honduras | Rural | Low (no road or market far) | State |
| Worku et al. (2014). [88] | Neutral/mixed | Ethiopia | Rural | Low (no road or market far) | |
| Yasuoka (2006). [89] | Neutral/mixed | Cameroon | Rural | Low (no road or market far) | |
| Becker (2000). [90] | Negative | Mali | Rural | Low | Mixed |
| Cavendish (2000). [91] | Negative | Zimbabwe | Rural | NR | Commons |
| Hue (2006). [92] | Negative | Vietnam | Rural | Good | Commons–private |
| Illukpitiya & Yanagida (2010). [93] | Negative | Sri Lanka | Rural | NR | NR |
| Jumbe & Angelsen (2006). [94] | Negative | Malawi | Rural | Mixed | NR |
| Koizumi et al. (2012) [95] | Negative | Indonesia | Rural | Low (no road or market far) | NR |
| Meaza & Demssie (2015) [96] | Negative | Ethiopia | Rural | Mixed | Mixed |

**Table 6.** *Cont.*

| Study | Gender Effect | Country | Rural/Urban Setting | Market Access | Forest Tenure |
|---|---|---|---|---|---|
| Moktan et al. (2009). [97] | Negative | Bhutan | Rural | NR | Commons |
| Mulenga et al. (2014). [98] | Negative | Zambia | Rural | NR | Commons |
| Mutenje et al. (2010). [99] | Negative | Zimbabwe | Rural | NR | Commons |
| Nielsen & Bakkegaard (2012). [100] | Negative | DRC | Rural | Very low (no road and market far) | State |
| Obua et al. (1998). [101] | Negative | Uganda | Rural | Good (road and market close) | State |
| Ocampo-Thomason (2006). [102] | Negative | Ecuador | Rural | NR | State |
| Paniagua-Zambrana et al. (2014). [103] | Negative | Colombia, Ecuador, Peru, and Bolivia | Rural | | State |

*3.10. Quality of the Evidence Base*

Most studies (53) did not satisfy the conditions of the Bilotta's risk of bias test. Of those that did satisfy the conditions, 17 were rated at low risk of bias, 6 were moderate risk of bias, and 1 high risk of bias. Guidance for the production of systematic maps suggests that critical appraisal is not necessary. We therefore took the decision to include all studies in the systematic map and not attempt any sub-analysis of the set of studies that were tested for potential bias. We note, however, that most studies were case studies, which are very difficult to generalise across sites. This creates substantial gaps in the evidence base.

**4. Key Findings**

This map sought to establish whether there is evidence that gender affects access to and use of forest assets for food security. After a rigorous assessment of the available evidence, 77 articles satisfied the agreed inclusion criteria. Of these included articles, 22 indicated a strong positive effect on gender, by reporting that women had a significantly better access to forests for food security. Food security was measured directly by reporting access to forest products for consumption or indirectly by income generation through sale of forest products. A large number of articles (41), however, reported mixed results, where the gender impact was both negative and positive for the same study. In such studies, men had better access to the commercial forest products, while women accessed products for subsistence consumption. A smaller number of studies (14) showed a clear negative impact on gender, where men had significantly better access to forest products for food security than women. In all studies across Africa, Asia, and Latin America, there is a strong gender specialisation, with commercial access and utilisation of forests and forest products being dominated by men, whereas access for subsistence and household consumption is almost exclusively the task of women.

**Supplementary Materials:** The following are available online at https://www.mdpi.com/article/10.3390/f12081096/s1, Table S1: Grey literature resources; Table S2: Test library of references for search comprehensiveness; Table S3: List of World Bank's low-and middle-income countries; Table S4: Search strings in Web of Science and Scopus and Simplified search string; Table S5: Data coding and extraction template; Critical appraisal template; Studies included on interactive systematic map.

**Author Contributions:** Conceptualization, N.K., J.C., L.C.-K., K.K. and G.P.; Data curation, N.K., B.S.-H. and G.P.; Formal analysis, N.K. and G.P.; Funding acquisition, N.K., L.C.-K. and G.P.; Investigation, N.K., B.S.-H.; Methodology, N.K., J.C. and G.P.; Supervision, J.C., L.C.-K. and G.P.; Validation, N.K. and G.P.; Visualization, G.P.; Writing—original draft, N.K., J.C., L.C.-K., K.K. and G.P.; Writing—review and editing, N.K. and G.P. All authors have read and agreed to the published version of the manuscript.

**Funding:** Funding was generously provided by a grant from the United Kingdom Department for International Development (DFID) through the Evidence Based Forestry Initiative at the Centre for International Forestry Research (CIFOR). The Swedish Ministry of Foreign Affairs special initiative for food security, through the Swedish University of Agricultural Sciences (SLU), supported the preparatory workshop on conducting systematic map protocols, as well as the participation of LCK. Additional funding for GP was provided by Oxford Systematic Reviews.

**Data Availability Statement:** Not applicable.

**Acknowledgments:** We thank Susanne von Walter for her invaluable input during the stakeholder workshop. We acknowledge, with gratitude and great sadness at her passing, the contribution of Esther Mwangi to the work of this project. We acknowledge William J. Harvey and Leo Petrokofsky of Oxford Systematic Reviews for help with the submission and preparation of the interactive systematic map, respectively. We thank two anonymous reviewers for their great help in improving the manuscript.

**Conflicts of Interest:** The authors declare that they have no conflict of interest.

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
