# Peer review of "What Is the Evidence Base Linking Gender with Access to Forests and Use of Forest Resources for Food Security in Low- and Middle-Income Countries? A Systematic Evidence Map"

_forests, doi:10.3390/f12081096_

Round 1

Reviewer 1 Report

This is a well written paper. The English is good and the data are presented in a clear way. Thus one can quickly read the paper and understand it.
It is a work with very carefully done statistics.
This is a useful addition to the literature.
But concrete data on forest resources for food security are missing.

Author Response

We thank the reviewer for their kind words.

We respond to the following comment below:

But concrete data on forest resources for food security are missing.

We have amended the subtitle under Figure 8 to "Forest resources and Food security, nutrition security and income". Data on what aspects of forest resources (including non-timber forest resources) are linked with food security are summarised in this sub-section and we have added details to provide more information. Data can also be viewed in the the full data sheet (Appendix 2) under column headings "food security outcome through engagement in: [forest harvesting, forest resource trading, forest resource processing, mix], and main ntfp assessed"

Reviewer 2 Report

The presented systematic map is an interesting contribution on how to inform the wider readership about the researched issues, which link gender with access to forests and use of forest resources.

I consider the methodological part, which is described in detail, to be important.

However, I also have a few comments that could help improve the manuscript.

I perceive the Background part as sufficient, but not very interesting for the readers.

The whole work is derived from the CEE methodology, which characterizes systematic review and systematic map. It would be useful to state why the systematic map approach was used, not the systematic review (although the whole article is referred to as the systematic review category).

Results: in the introduction, the results refer to the interactive map on the page https://oxsrev.github.io/evidencemaps/gender/ but this page is not available. Some parts (e.g. Search results including Figure 1 or the first paragraph of the chapter Narrative synthesis) should be mentioned in the methodology rather than in the results. Most of the results are presented in the form of a graph or figures and tables (which is a suitable form intended for the professional public), for a scientific article it would be more appropriate to use a more detailed written description. Some results are therefore a bit misleading, e.g. inserting a trend curve into Figure 2 is inappropriate (especially because the graph does not include years with zero incidence 1986-1994, 1999). It is clear that the purpose of systematic maps is not an in-depth analysis of relationships, yet some of the selected analyzed criteria seem unnecessary. An example is altitude rating. On the contrary, if you evaluate the authorship of the article according to the Location of first author’s Institution and at the same time, the work is focused on the issue of gender, I would expect, for example, a gender analysis of the first authors. Table 5 evaluates the Gender effect, it is not entirely clear how it was evaluated (or what the Positive / Negative effect means - it would be appropriate to document by an example).

The introduction presents research questions. These are set logically. However, if the questions were set, I would expect them to be answered directly in the final chapter. On the contrary, the only sentence in the key findings is insufficient. If this passage will be not extended, it would be more appropriate to withdraw it altogether.

Author Response

We thank the reviewer for their kind words about the paper and for their helpful comments to improve the manuscript. Details of our responses are listed below:

1. I perceive the Background part as sufficient, but not very interesting for the readers.

We have substantially reduced this section as much of what we consider to be the Background to the review is set out in the equivalent section in the Protocol. Owing to an oversight, the reference to this Protocol was missing from our submission and we apologise to the Reviewer, therefore, that this important link was missing. We have not therefore amended the Background section - we are not clear precisely how to make it 'more interesting'. but we hope that the reference to our earlier work will provide the necessary additional information.

2. The whole work is derived from the CEE methodology, which characterizes systematic review and systematic map. It would be useful to state why the systematic map approach was used, not the systematic review (although the whole article is referred to as the systematic review category).

We have clarified why we adopted a systematic mapping approach in the manuscript. We followed reporting guidelines set out in ROSES RepOrting standards for Systematic Evidence Syntheses (Haddaway et al, 2018 - 

https://doi.org/10.1186/s13750-018-0121-7)

in referring to "review" throughout, as being clearer than "map" and less cumbersome than "systematic map".

3. Results: in the introduction, the results refer to the interactive map on the page https://oxsrev.github.io/evidencemaps/gender/ but this page is not available.

We apologise to the reviewer if the link to the interactive map was not working when they were reviewing the paper owing to an error in the link provided. This is an important component of our work and we acknowledge that without it the manuscript is incomplete. We have edited the url and the link is working, allowing readers to interact with the systematic map in full.

4. Some parts (e.g. Search results including Figure 1 or the first paragraph of the chapter Narrative synthesis) should be mentioned in the methodology rather than in the results.

We have added details about preparing the interactive map to the method section. We followed ROSES guidelines on reporting (see ref above) and the flow chart of article selection is considered a major Result, we have therefore left Figure 1 in the Result section.

5. Most of the results are presented in the form of a graph or figures and tables (which is a suitable form intended for the professional public), for a scientific article it would be more appropriate to use a more detailed written description.

We feel that it there is sufficient written description in the manuscript as it stands. Adding further written description to clear graphs and figures would add bulk to the paper but not make the results clearer, except in specific sections referred to in response to specific review comments.

6. Some results are therefore a bit misleading, e.g. inserting a trend curve into Figure 2 is inappropriate (especially because the graph does not include years with zero incidence 1986-1994, 1999).

We agree that the trend line is inappropriate and have removed it. We have amended a number of other charts to improve their quality and consistency with other charts in the paper.

7. It is clear that the purpose of systematic maps is not an in-depth analysis of relationships, yet some of the selected analyzed criteria seem unnecessary. An example is altitude rating.

We developed the extraction strategy (i.e. criteria to extract and analyse) with stakeholders and agreed it in the published Protocol. It is true that once we examined all the included papers, there was not always a great deal to report for some categories, but for completeness sake, and in accordance with Guidelines for systematic evidence syntheses, we reported all the categories that we agreed to assess in the Protocol, including altitude of the study sites.

8. On the contrary, if you evaluate the authorship of the article according to the Location of first author’s Institution and at the same time, the work is focused on the issue of gender, I would expect, for example, a gender analysis of the first authors.

We do not think that it is possible to report on the gender of authors without sending them a questionnaire. While this may be an interesting aspect to study, it was not considered a priority by our stakeholders in terms of data to code and extract and report on. However, the location of the principal research institution does reveal that, in common with most development studies, research about aspects of low- and middle-income experiences are carries out in developed country research institutions. We do not make further comments about this, but the data can be interpreted by readers if they wish. It is data that could potentially contribute to wider studies of such development research and the uneven distribution of science publishing in developing countries and emerging economies.

9. Table 5 evaluates the Gender effect, it is not entirely clear how it was evaluated (or what the Positive / Negative effect means - it would be appropriate to document by an example).

We have added details of how this was undertaken and added the field of textual comments extracted from included papers to the data table and to the interactive map. It is now possible to filter the map according to the categories positive, neutral and negative. We include caveats about over-interpreting these categories.

10. The introduction presents research questions. These are set logically. However, if the questions were set, I would expect them to be answered directly in the final chapter. On the contrary, the only sentence in the key findings is insufficient. If this passage will be not extended, it would be more appropriate to withdraw it altogether.

We have added text to draw the research questions together in our Conclusions and key findings. We thank the reviewer for this valuable suggestion and preferred not to withdraw the original key findings but to supplement them, as suggested.

Round 2

Reviewer 2 Report

Thanks to the authors for improving the manuscript. 

All my main objections and comments have been taken into account.